# SIO: A Spatioimageomics Pipeline to Identify Prognostic Biomarkers Associated with the Ovarian Tumor Microenvironment

**DOI:** 10.3390/cancers13081777

**Published:** 2021-04-08

**Authors:** Ying Zhu, Sammy Ferri-Borgogno, Jianting Sheng, Tsz-Lun Yeung, Jared K. Burks, Paola Cappello, Amir A. Jazaeri, Jae-Hoon Kim, Gwan Hee Han, Michael J. Birrer, Samuel C. Mok, Stephen T. C. Wong

**Affiliations:** 1Center for Modeling Cancer Development, Houston Methodist Cancer Center, Houston Methodist Hospital, Houston, TX 77030, USA; yzhu@houstonmethodist.org (Y.Z.); jsheng@houstonmethodist.org (J.S.); 2Departments of Pathology and Laboratory Medicine and Radiology, Houston Methodist Hospital, Weill Cornell Medicine, Houston, TX 77030, USA; 3Department of Gynecologic Oncology and Reproductive Medicine, The University of Texas MD Anderson Cancer Center, Houston, TX 77030, USA; sferri@mdanderson.org (S.F.-B.); tszlun.yeung701@gmail.com (T.-L.Y.); AAJazaeri@mdanderson.org (A.A.J.); 4Department of Leukemia, The University of Texas MD Anderson Cancer Center, Houston, TX 77030, USA; jburks@mdanderson.org; 5Department of Molecular Biotechnology and Health Sciences, University of Turin, 10126 Turin, Italy; paola.cappello@unito.it; 6Department of Obstetrics and Gynecology, Yonsei University College of Medicine, Seoul 03722, Korea; jaehoonkim@yuhs.ac (J.-H.K.); lachica@yuhs.ac (G.H.H.); 7Winthrop P. Rockefeller Cancer Institute, The University of Arkansas for Medical Sciences, Little Rock, AR 72205, USA; MJBirrer@uams.edu

**Keywords:** cancer microenvironment, imaging mass cytometry, deep learning, transcriptomic profiling, high-grade serous ovarian cancer, tumor biomarkers, survival prediction

## Abstract

**Simple Summary:**

High-grade serous ovarian cancer (HGSC) caused more than 13,000 deaths annually in the United States. A critically important component that influences the HGSC patient survival is the tumor microenvironment. However, how different cells interact to influence HGSC patients’ survival remains largely unknown. To investigate this, we developed a pipeline that combines imaging mass cytometry (IMC), location-specific transcriptomics, and deep learning to identify the distribution of various stromal, tumor and immune cells as well as their spatial relationship. Our pipeline automatically and accurately segments cells and extracts salient cellular features to identify biomarkers, and multiple nearest-neighbor interactions among different cells that coordinate to influence overall survival rates in HGSC patients. In addition, we integrated IMC data with microdissected tumor and stromal transcriptomes to identify novel signaling networks. These results may lead to the discovery of novel survival rate-modulating mechanisms in HGSC patients.

**Abstract:**

Stromal and immune cells in the tumor microenvironment (TME) have been shown to directly affect high-grade serous ovarian cancer (HGSC) malignant phenotypes, however, how these cells interact to influence HGSC patients’ survival remains largely unknown. To investigate the cell-cell communication in such a complex TME, we developed a SpatioImageOmics (SIO) pipeline that combines imaging mass cytometry (IMC), location-specific transcriptomics, and deep learning to identify the distribution of various stromal, tumor and immune cells as well as their spatial relationship in TME. The SIO pipeline automatically and accurately segments cells and extracts salient cellular features to identify biomarkers, and multiple nearest-neighbor interactions among tumor, immune, and stromal cells that coordinate to influence overall survival rates in HGSC patients. In addition, SIO integrates IMC data with microdissected tumor and stromal transcriptomes from the same patients to identify novel signaling networks, which would lead to the discovery of novel survival rate-modulating mechanisms in HGSC patients.

## 1. Introduction

Advanced high-grade serous ovarian cancer (HGSC) is the most lethal gynecologic malignancy, causing more than 13,000 deaths annually in the United States [1]. HGSC is notable for initial sensitivity (75% response rate) to platinum and taxane neoadjuvant chemotherapy or chemotherapy following debulking surgery [2,3]. However, most tumors (>75–80%) recur within 12 to 24 months after treatment, and many patients die of progressively chemotherapy-resistant disease [4,5,6]. A critically important component that influences the patient survival is the tumor microenvironment [7,8], which is primarily composed of fibroblasts, extracellular matrix proteins, endothelial cells, lymphocytic infiltrates, and cancer cells. The tumor microenvironment has been shown to directly affect cancer cell growth, migration, invasion, chemoresistance, cell-cell interactions, and matrix remodeling [9,10]. However, spatially resolved, single-cell analysis that can identify tumor and stromal cell phenotypes, characterize their heterogeneity and cell-cell interactions, and biomarkers for predicting survival of HGSC patients, are lacking.

Several approaches have been used to perform spatial analysis of the tumor microenvironment. Distance analysis associated cell-cell spatial distance with clinical outcomes in different types of cancer (breast [11], gastric [12]). Spatial statistics that used L-function or K-function to detect deviation from spatial homogeneity has been employed to study architectural patterns of cells [13,14]. Neighborhood analysis that generated cell social interaction network and clustered images based on significant cell-cell interactions was applied to analyze highly multiplexed mass cytometry images [15]. Spatial community analysis identified cell communities on the basis of physical proximity and associated densities of cell communities with clinical outcome of breast cancer [16]. Nevertheless, all these methods do not support automated extraction of spatial features, specifically cell-cell interactions, that can have prognostic values as captured by highly multiplexed tumor images.

Imaging mass cytometry (IMC) is an imaging-based mass cytometry (CyTOF) that couples immunohistochemical and immunocytochemical methods with high-resolution laser ablation [17] to allow the imaging of more than 30 proteins and protein modifications simultaneously at subcellular resolution. This enables researchers to uncover the heterogeneity of cellular phenotypes and cell-cell interactions (phenograph [18]; histocat [15]). Cell segmentation is the first key step of IMC analysis. However, IMC images of biological tissues, particularly those of solid tumors, are extremely challenging for conventional cell segmentation methods, such as the watershed algorithm [17] and pixel-based classification [15], owing to great variations in image intensities and cell shapes, overlapping cells, dense cell clusters, blurred edge information, missing object borders, and low signal-to-noise ratios. In addition, the highly multiplexed IMC data generate rich information of cell phenotypes, spatial organization, and heterogeneity. However, methods to quantify and integrate various types of data in order to identify reliable prognostic biomarkers remain largely unexplored.

Mask Region-based Convolutional Neural Network (Mask-R-CNN or MRCNN), is an advanced deep learning method that adopts a two-stage procedure, with a Region Proposal Network (RPN) in the first stage and a parallel prediction of the class, the box offset and a binary mask for each ROI in the second stage [19]. MRCNN was initially designed for object detection and instance segmentation of natural images [19]. It outperformed all existing single-model entries on every task in the recent Microsoft Common Objects in Context (COCO) challenge, one of the most authoritative competitions in object detection and segmentation [20]. MRCNN was also adapted to perform nuclei segmentation in histologic microscopic images [21]. However, whether MRCNN could be employed in the IMC cell segmentation has not been investigated yet.

In our study, we built a machine learning enabled SIO pipeline that integrates, processes, models, and analyzes highly multiplexed, subcellular IMC data and transcriptomic data generated from well-annotated and treatment-naïve HGSC samples. We adapted MRCNN for IMC cell segmentation and used logistic regression to identify salient prognostic spatial features of cell-cell interactions for predicting patient survival rates by using spatial information provided by IMC. We combined the quantitated IMC images with cancer and stromal gene expression data of microdissected tissue specimens from the same HGSC patients to detect genes that are significantly correlated with prognostic features. We filtered tumor or stroma/fibroblastic specific genes with single-cell RNA seq data [22] and postulated new mechanisms by which these genes contribute to the prognostic features of cancer survival prediction.

## 2. Materials and Methods

### 2.1. Patient Samples

A total of 41 paraffin-embedded tumor tissue samples obtained from patients with advanced stage (stage IIIB-IV) high-grade serous ovarian cancer (HGSC) were used in the study. Tissue samples were obtained from the ovarian cancer repositories at The University of Texas MD Anderson Cancer Center in Houston, Texas and Gangnam Severance Hospital, Yonsei University College of Medicine in Seoul, South Korea. They were collected from previously untreated patients undergoing primary cytoreductive surgery for ovarian cancer. After surgery, patients received platinum-based combination chemotherapy. Optimal surgical cytoreduction was defined by a residual tumor no more than 1 cm in diameter. The overall survival duration was measured from the date of diagnosis to the date of death or censored at the date of the last follow-up examination. Long-term survivors were those with an overall survival time ≥ 60 months while short-term survivors were those with an overall survival time ≤ 20 months. Clinical data, including age, cytoreduction status (optimal vs. suboptimal), and overall survival, were obtained from the records of the patients with HGSC. All samples and clinical data were collected with the approval of the Institutional Review Boards of MD Anderson and Gangnam Severance Hospital.

### 2.2. Preparation and Staining

Tissue slides were deparaffinized in xylene followed by rehydration in a graded alcohol series. Antigen retrieval was performed with citrate buffer (pH 6) at 95 °C in a decloaking chamber (Biocare Medical, Pacheco, CA, USA) for 25 min. Slides were then blocked with 3% bovine serum albumin in phosphate-buffered saline for 30 min and incubated for 2 h at room temperature with 21 metal-tagged antibodies (Appendix A). Following incubation, tissue slides were washed with phosphate-buffered saline and incubated with 0.5 µM Cell-ID Intercalator-Ir (Fluidigm, South San Francisco, CA, USA) for the detection of nuclear DNA. Slides were then rinsed in phosphate-buffered saline and air-dried.

### 2.3. Imaging Mass Cytometry

Imaging mass cytometry (IMC) data were acquired by a Fluidigm Helios CyTOF instrument equipped with a Hyperion System laser ablation module in the Flow Cytometry and Cellular Imaging Facility at MD Anderson. A total of 41 images of 1 mm^3^ each were acquired and used for the current study, including 20 images from short-term survivors (overall survival ≤ 20 months) and 21 images from long-term survivors (overall survival ≥ 60 months). Each 1 mm^2^ region of interest on the tissue section was selected based on the image from the corresponding hematoxylin and eosin stained serial tissue section, which demonstrated representative of tumor regions surrounded by stomal cells.

### 2.4. Microdissection and Microarray Analysis of Tissue Samples

RNA was extracted from microdissected frozen HGSC samples, which included tumor epithelial components and stromal components from sixteen HGSC patients, from whom IMC data were available. Extensive details of specimen handling, RNA extraction and amplification, microarray hybridization, and quality-control procedures have been described previously [23].

### 2.5. Data Preprocessing and Cell Segmentation

Data were converted to TIFF format by MCD viewer (Fluidigm). Channel spillover was compensated using the nonnegative least square approach [24,25]. Watershed segmentation was performed on the maximal projection of normalized images of H3 and nucleic acid intercalator (^191^Ir and ^193^Ir). The best Watershed segmentation results were achieved with prior median filtering (2 × 2 pixels) followed by Gaussian blurring (kernel width of 2 pixels), and standard parameters for watersheds. These steps were performed by an in-house developed Matlab script with the Matlab image processing toolbox. Mask-R-CNN (MRCNN) segmentation was trained on the outputs of Watershed segmentation. Thirty-one images were employed for training, with each image segmented into 16 small pieces to increase the training speed (each small image had a size of ~250 × 250 pixels, image resolution 1 μm/pixel). A total of 496 small images were used as the training set. We applied an MRCNN model with a feature pyramid network and a convolutional neural network ResNet-101 backbone based on an implementation (https://github.com/matterport/Mask_RCNN) by Matterport Inc. (Sunnyvale, CA, USA, released under an MIT License, accessed on: 3 May 2019) that employed the python open-source libraries Keras and Tensorflow. MRCNN has three outputs for each candidate object, a class label, a bounding-box offset, and the object mask [19]. During training, MRCNN uses a multi-task loss on each sampled RoI as L = L_cls_ + L_box_ + L_mask_ where L_cls_ is classification loss, L_box_ is bounding-box loss, and L_mask_ is the average binary cross-entropy loss [19]. We initiated the model using weights obtained from pretraining on the MSCOCO dataset [20]. We started with the learning rate of 0.001 and trained with 50 epochs and decreased the learning rate to 0.0001 and trained for 50 epochs. The training was performed using one NVIDIA V100 GPU (Amazon AWS p3.2xlarge instance). To compare the results of Watershed and MRCNN segmentation, we manually segmented ten testing images (~300 cells per image, each image had a size of ~250 μm × 250 μm) to be used as a reference segmentation. To quantify the performance of different segmentation methods, we computed the Sørensen–Dice coefficient, $\frac{2\left|x⋂y\right|}{\left|x\right|+\left|y\right|}$, between each cell in Watershed segmentation and its maximum overlapping cell in reference segmentation, and between each cell in MRCNN segmentation and its maximum overlapping cell in reference segmentation. We compared the mean and standard deviation of the Sørensen–Dice coefficient of all the cells in each sample between Watershed and MRCNN segmentations.

### 2.6. Analysis Workflow

MRCNN segmentation, survival prediction, and correlation of cell density with gene expression were implemented in python 3.5. Watershed segmentation, image analysis, and the single-cell analysis algorithms were performed by Matlab R2016a. Phenograph clusterings and heatmap figures for cell subtype annotation were generated by R 3.6.

### 2.7. Clustering Analysis

The mean intensity of each marker j within each cell k of an image m, I_jk_^m^ was calculated as the mean intensity of all the pixels within the segmentation of that cell. It was normalized by calculating the z-score z_jk_^m^ = (I_jk_^m^ − μ_j_^m^)/σ_j_^m^, where μ_j_^m^ and σ_j_^m^ are the mean and standard deviation of I_jk_^m^ for k = 1, 2, 3,…, N^m^ in the image m for that marker j, and N^m^ is the total number of cells in the image m. Phenograph clustering was performed using the Matlab cyt package. In the first step, the normalized data were under-clustered to detect and separate the major cell populations using 100 nearest neighbors. Sixteen markers were used: SMA, CD14, CD163, CD11b, CD45, CD44, CD4, CD73, CD68, CD20, CD8a, granzyme B, Ki67, Coll-I, CD45RO, and Keratin8/18. Of the nineteen clusters generated, nine tumor or stroma clusters were kept, and 10 non-tumor and non-stroma clusters were pooled together and underwent a second round of clustering using 80 nearest neighbors and 15 markers: CD14, CD163, CD11b, CD31, CD45, CD44, CD4, CD73, CD68, CD20, CD8a, CD196, granzyme B, Ki67, and CD45RO. Of the eighteen clusters generated, three macrophage clusters were pooled together and were further clustered by markers: CD44, CD14, CD163, CD68, CD4, CD45RO, and CD11b. Similarly, CD8^+^ T cell cluster, CD4^+^ T cell cluster, and clusters that had mixed tumor and immune cells underwent further clustering using their related markers. Finally, all the clusters were gathered together and visualized in Barnes-Hut t-SNE [26], a two-dimensional representation of high dimensional data.

### 2.8. Cell Density and Nearest-Neighbor Interactions in Tumor-Enriched Regions

Tumor-enriched regions were calculated as a thresholded two-dimensional Gaussian convolved image (σ = 15 pixels, threshold = 0.0005) of the density map of the centroid of tumor cells. The regions outside of tumor-enriched regions were defined as tumor-unenriched regions. Cell density in tumor-enriched regions was computed as the cell count in the tumor-enriched region per total tumor cells. For the nearest-neighbor cell-cell interactions (if the distance between the centroids of two cells < 20 μm, the two cells are considered as the nearest neighbors), the average cell count Avg.C_m_^j^ of any cell subtype j in the nearest neighborhood of each cell subtype of interest m in the tumor-enriched region was computed as Avg.C_m_^j^ = N_mj_/N_m_, where N_mj_ is the total number of nearest neighboring pairs of cell subtypes m and j of each sample, and N_m_ is the total cell counts of cell subtype m in the tumor-enriched region of each sample. An unpaired *t* test was used to determine if there was a significant difference (adjusted *p* < 0.05) between the means of nearest-neighbor cell-cell interactions of long-term and short-term survivors. Adjustment for multiple testing was conducted using the Benjamini-Hochberg method [27].

### 2.9. Survival Prediction

The training and test data sets were randomly split at 80% and 20% of the total 41 samples containing 21 long-term survivors and 20 short-term survivors. The ratio of the long-term to short-term survivors in the training data set was 1 while in the test data set was 1.25. For survival prediction using only cell density and age, the forty-one cell density and age features were first subjected to Spearman correlation with survival, and 22 features that had an absolute correlation coefficient larger than 0.2 were kept. Since logistic regression assumes independence between features, features could not be highly correlated. Only one of the highly correlated features (absolute Spearman correlation coefficient ≥ 0.65) that had the highest correlation with survival was selected and the rest were dropped, leaving 17 features. Each feature was normalized to a 0 to 1 scale. Recursive feature elimination and logistic regression (Python’s sklearn package) were used to rank the features according to their importance. Owing to the small sample size, leave-one-out cross validation was used to evaluate performance during the training. The optimal feature number was selected at the highest validation accuracy.

To narrow down the nearest-neighbor interaction features that were related to the prognostic cell density features, we selected forty-six features of the nearest-neighbor interaction that contained any of the seven prognostic cell density features and were also significantly different between long-term and short-term survivors and combined them with age to predict survival. To filter the features, we performed Spearman correlation between survival and these forty-seven features. Forty-four features that had an absolute correlation coefficient larger than 0.2 were kept. Because features of logistic regression cannot be highly correlated, only one of the highly correlated features (absolute Spearman correlation coefficient ≥ 0.65) that had the highest correlation with survival was kept, leaving 26 features. Each feature was normalized to a 0 to 1 scale. Recursive feature elimination and logistic regression were used to rank the features according to their importance. Owing to the small sample size, leave-one-out cross validation was used to evaluate performance during the training. The optimal feature number was selected at the highest validation accuracy.

### 2.10. Correlation of Cell Density with Gene Expression

Gene expression was normalized by Robust Multi-array Average [28]. Spearman correlation between gene expressions in microdissected stromal or epithelial components and cell densities in tumor-enriched regions was performed and genes of interest were selected from those that had positive correlation coefficients (*p* < 0.05, absolute correlation coefficient > 0.4) with each cell subtype that showed a significant difference between long-term survivors and short-term survivors. For genes that had multiple probe IDs, we used only the probe ID that had the largest variance. The genes of interest of the microdissected stromal components were filtered by the single-cell RNA sequencing (scRNAseq) data [22], and only the genes expressed in fibroblasts or stromal cells were kept. Moreover, to understand the molecular mechanism that might explain the correlation between IMC and HGSC patient survival, we included only genes encoding for secreted or receptor proteins for all listed IMC features, except for CD73_1 and CD73_2 (for which we included all genes). The genes of interest of the microdissected epithelial components were filtered by the scRNAseq data and only genes expressed in epithelial cells were retained. Among these, genes encoding for secreted or receptor proteins for all listed IMC features, except for tu_9 (for which we included all genes), were included. Kaplan-Meier analysis was performed on the filtered genes of interest, and only genes that had significant prognostic values (*p* < 0.05) were retained.

### 2.11. Kaplan-Meier Analysis

Kaplan-Meier analysis was performed differently for genes in stromal and epithelial components. For genes in stromal components, Kaplan-Meier analysis was performed on our microdissected gene expression data in the stromal component (70 patients, overall survival < 150 months). For genes in epithelial components, Kaplan-Meier analysis was performed using an online tool, KMplotter [29], which employs a database of gene expression data and survival information of 530 HGSC patients (stage II/III/IV; grade 2/3; optimal debulking), downloaded from Gene Expression Omnibus and the Cancer Genome Atlas. To analyze the prognostic value of each selected gene, we divided the patients into two groups according to various quantile expressions of the gene, and the best performing threshold was used as a cutoff. Statistical comparison of the two groups was performed using the log-rank test.

### 2.12. Data and Code Availability

Raw data of the microdissected transcriptomes were downloaded from GEO with accession number GSE115635. The datasets/code supporting the current study are available from https://data.mendeley.com/datasets/4dpk7fjb58/draft?a=4aa42b24-f791-4a89-b74e-9dba795f3755, accessed on: 30 March 2021.

## 3. Results

### 3.1. Image Analysis Pipeline

To comprehensively quantify the cellular heterogeneity and spatial organization of HGSC tissue and find biomarkers that predict patient survival, we used IMC to detect 21 different proteins in 41 tumor samples from treatment-naïve HGSC patients (Appendix A). Tissue sections were stained with a panel of metal-tagged antibodies (Appendix A) followed by laser ablation coupled to mass spectrometry to generate high-dimensional images as previously described [11] (Figure 1). Our selected panel consisted of markers of proliferation; immune cell regulators; and markers of epithelial, stroma, immune and endothelial lineages (Appendix A).

Resulting data were then analyzed using a novel IMC-adapted image analysis pipeline. Briefly, cell segmentation was performed using the deep learning method, MRCNN, followed by phenograph clusterings to identify and annotate different cell subtypes used for cell density and neighborhood analyses. These two features were then used for survival prediction analysis and correlating IMC phenotype with gene expression profile (Figure 1).

### 3.2. Cell Segmentation and Annotation by Deep Learning-Based IMC Data Analysis

We employed MRCNN for IMC cell segmentation. We used the computerized outputs of watershed segmentation as the training sets for MRCNN. The mean Sørensen–Dice coefficient of MRCNN was significantly higher than that of watershed (mean of mean Sørensen–Dice coefficient of MRCNN = 0.71, mean of mean Sørensen–Dice coefficient of watershed = 0.64, *p* = 1.46 × 10^−6^, *n* = 10 samples, paired *t* test; Figure 2B), and the standard deviation of Sørensen–Dice coefficient of MRCNN was significantly lower than that of watershed (mean standard deviation of Sørensen–Dice coefficient of MRCNN = 0.17, mean standard deviation of Sørensen–Dice coefficient of watershed = 0.25, *p* = 9.3 × 10^−6^, *n* = 10 samples, paired *t* test; Figure 2C). These results suggest that the segmentation by MRCNN is more similar to the manual segmentation than the watershed segmentation, and that it has less over-segmentation issue than watershed (Figure 2A and Figure 3A).

After performing phenograph clusterings to identify cell subtypes, we identified 40 cell subtypes of T and B cell, macrophage, endothelial, and other stroma cell populations as well as tumor cell subtypes from 162,869 cells in 41 images (Figure 3B) and quantified the normalized expression of all markers across various cell subtypes (Figure 3C). Specifically, based on the normalized expression of cell subtype–specific markers, we identified nine macrophage/monocyte subtypes (Figure 4A), four CD8^+^ T cell subtypes and 6 CD4^+^ T cell subtypes (Figure 4B), nine tumor subtypes (Figure 4C), and many other stroma and immune cell subtypes (Figure 4D). Since the normalization step converts the expression of all cells in a sample to a z-score (see Methods), the normalized expression reflects the relative expression across all cells. For example, although tumor cells subtypes (such as tu_1, tu_2, tu_3, tu_5, tu_6, tu_8, and tu_9) have median normalized Keratin 8/18 expression around zero (Figure 4C), they exhibit significantly higher Keratin 8_18 expression levels than non-tumor cells (Appendix A). Based on the results of Figure 3C and Figure 4, the phenotypes of all cell subtypes are summarized in Figure 4. The spatial distribution of all cell subtypes in one representative sample can be visualized in Appendix A.

### 3.3. Spatially Resolved Cell Density and Nearest-Neighbor Cell-Cell Interactions Analyses of the Ovarian Tumor Microenvironment

We automatically calculated the tumor-enriched region for each image (see Methods). By computing the cell density as the cell count in the tumor-enriched region per total number of tumor cells, we found several cell subtypes exhibiting significant differences between long-term survivors (LTS; overall survival ≥ 60 months, *n* = 21) and short-term survivors (STS; overall survival ≤ 20 months, *n* = 20; Figure 5A). Among different T cell subtypes, granzyme B^+^ CD8^+^ cytotoxic T cell (CD8_4) density was significantly higher in LTS (*p* = 0.019, Figure 5A) and CD45RO^+^ CD44^+^ CD8^+^ memory T cell (CD8_3) density had a declining trend of LTS (*p* = 0.084, Appendix A) than of STS. In addition, CD45RO^+^ CD4^+^ memory T cell (CD4_4) density was significantly higher in LTS than in STS (*p* = 0.024, Figure 5A). Among different CD73^+^ cell subtypes, CD73^+^ cell (CD73_1) density and CD73^mid^ cell (CD73_2) density were significantly lower in the LTS than in STS (*p* = 0.015 and *p* = 0.002, respectively; Figure 5A). CD31^+^ CD73^mid^ endothelial cell (CD31) density was significantly lower in LTS than in STS (*p* = 0.007, Figure 5A). Among tumor cell subtypes, B7H4^+^ Keratin^+^ tumor cell (tu_9) density was significantly lower in LTS than in STS (*p* = 0.018, Figure 5A). A comparison of cell densities of all cell subtypes between LTS and STS is shown in Appendix A.

To determine whether the appearance of one cell subtype is associated with the appearance of another cell subtype, a Spearman correlation matrix between various cell subtype densities was generated. The results demonstrated that the granzyme B^+^ CD8^+^ cytotoxic T cell (CD8_4) density was negatively correlated with cell densities of B7H4^+^ tumor cells (tu_7; r = −0.47, *p* = 0.0006) and tu_9 (r = −0.38, *p* = 0.006; Figure 5B), suggesting that these CD8^+^ cytotoxic cells are infiltrating the tumor mass and actively depleting B7H4^+^ ovarian cancer cells in LTS. CD163^+^ CD68^+^ CD14^+^ macrophage (ma_3) density was positively correlated with CD45RO^+^ CD44^+^ CD4^+^ memory T cell (CD4_1) density (r = 0.56, *p* = 2.8 × 10^−5^), CD45RO^mid^ CD44^+^ CD4^+^ memory T cell (CD4_3) density (r = 0.56, *p* = 2.4 × 10^−5^), and CD45RO^mid^ CD44^mid^ CD4^+^ memory T cell (CD4_5) density (r = 0.45, *p* = 0.001) (Figure 5B). A simultaneous high expression of CD163 and CD68 has been used as a marker for M2 macrophage [30,31]. However, previous studies demonstrated that tumor-associated macrophages are diverse and heterogeneous, and do not have restricted M1 or M2 phenotypes [32,33]. Our results indicated a positive correlation between CD163^+^ CD68^+^ CD14^+^ cell type (ma_3) and CD4^+^ memory T cells (CD4_1, CD_3, CD_5) (Figure 5B).

Next, we examined the prognostic significance of nearest-neighbor cell-cell interactions by computing the average cell count (Avg. C) of any cell subtype in the nearest neighborhood of each cell subtype of interest (distance between the center of two cells less than 20 μm) in the tumor-enriched region of every LTS or STS patient sample. We computed the nearest-neighbor cell-cell interactions that were significantly higher or lower (Benjamini-Hochberg adjusted *p* value < 0.05) in LTS (*n* = 21) than in STS (*n* = 20). We identified 120 nearest-neighbor cell-cell interactions that were significantly different between LTS and STS (Figure 6A). Among them, granzyme B^+^ CD8^+^ cytotoxic T cells (CD8_4) had significantly more interactions with multiple tumor cell subtypes (tu_1, tu_2, tu_3, and tu_5) in LTS than in STS (Figure 6A), suggesting increased interactions between CD8 cytotoxic cells and multiple subtypes of tumor cells in LTS. An example of the interaction between CD8_4 and tu_1 is shown in Appendix A.

In contrast to more CD8_4–tumor cell interactions in LTS than in STS, CD73^mid^ cell (CD73_2) had significantly fewer interactions with 17 cell subtypes, including CD73_1, CD31, macrophages and monocytes (ma_1, ma_2, ma_4, ma_5, ma_8, ma_9), stromal cells (s_1, s_2), T cells (CD4_5, CD8_2), and tumor cells (tu_1, tu_3, tu_4, tu_5, tu_6), than in STS, suggesting that when there are more CD73_2 cells in the tumor microenvironment, most of them are surrounded by macrophages and tumor cells in STS. The interaction between CD73_2 and CD163^+^ CD68^+^ CD14^+^ macrophages (ma_9) is shown in Appendix A. CD4_4 cells had significantly more interactions with CD163^+^ CD68^+^ Vista^mid^ CD14^+^ macrophages (ma_1; Appendix A) and CD14^+^ monocytes (ma_2) in LTS than in STS (Figure 6A), suggesting increased interactions between CD4^+^ memory T cells and certain subtypes of macrophages in LTS. CD45RO^+^ CD44^+^ CD8^+^ memory T cells (CD8_3) had significantly fewer interactions with CD163^+^ CD14^mid^ macrophages (ma_8; Appendix A, Figure 6A) in LTS than in STS, suggesting decreased interactions between CD8^+^ memory T cells and this subtype of macrophages in LTS.

### 3.4. Feature Selection for Overall Survival Prediction by Logistic Regression

We deployed a common machine learning method, logistic regression, to identify prognostic features that predict survival. After splitting the data (41 samples) into training (80%) and test (20%) sets, we filtered out highly correlated features and kept the ones that had the highest correlation with survival. We next performed recursive feature elimination (RFE) on these features (see Methods). Our first approach used only the cell densities detected in tumor-enriched regions and patient age as features (Figure 6B). The optimal number of selected features was seven, because both training accuracy (0.947) and validation accuracy (0.938) were high (Figure 6B, top). The test accuracy was 0.78 (test sensitivity = 0.6, test specificity = 1) and the area under the curve (AUC) was 0.8 (Figure 6B, middle). Among the seven prognostic features selected by the model, CD73^mid^ cells (CD73_2), CD31^+^ CD73^mid^ endothelial cells (CD31), CD163^+^ CD68^+^ Vista^mid^ CD14^+^ macrophages (ma_1), CD45RO^+^ CD44^+^ CD8^+^ memory T cells (CD8_3), and age had negative coefficients, suggesting that they were inversely correlated with patient survival. Granzyme B^+^ CD8^+^ cytotoxic T cells (CD8_4) and CD45RO^+^ CD4^+^ memory T cells (CD4_4) had positive coefficients, suggesting that they were positively correlated with patient survival (Figure 6B, bottom, Figure 5A).

Next, we used the nearest-neighbor, cell-cell interactions in the tumor-enriched region, which are related to the seven prognostic cell density features in Figure 6B, bottom, and patient age as features (Figure 6C). The optimal number of selected features was 11 as both training accuracy (1) and validation accuracy (1) were the highest for that number of features (Figure 6C, top). The test accuracy was 0.89 (test sensitivity = 1, test specificity = 0.75) and the AUC was 1 (Figure 6C, middle). Among the eleven features selected by the logistic regression model, average cell count (Avg.C) of CD73_2 neighboring CD73_1, ma_9, or s_2; CD31 neighboring tu_4; CD8_3 neighboring ma_8; and age were negatively correlated with patient survival. In contrast, Avg.C of CD4_4 neighboring s_2, s_1, ma_1, or CD44 and CD8_4 neighboring tu_1 was positively correlated with survival (Figure 6C, bottom). Some of the prognostic nearest-neighbor interaction features, such as Avg.C of CD8_4 neighboring tu_1, CD73_2 neighboring ma_9, CD4_4 neighboring ma_1, and CD8_3 neighboring ma_8, can be visualized in Appendix A. These results indicate that multiple neighboring interactions between stromal, immune, and tumor cells may work together to influence patient survival. The complex cell-cell interaction patterns of ovarian cancer with various immune cell and other stromal subtypes led to divergence in the tumor microenvironment between LTS and STS.

Taken together, our results indicated that using nearest-neighbor, cell-cell interactions and age as features allowed a more accurate prediction of patient survival than using cell densities and age as features. The Spearman correlation of any two nearest-neighbor interaction features that both had relatively high correlation with patient survival (Spearman correlation coefficient > 0.2) and were related to the seven prognostic cell density features were visualized in Figure 6D. The Spearman correlation study demonstrated that certain features were highly correlated. For example, we found that the average cell count (Avg.C) of CD8_4 neighboring with a tu_1 cell was highly correlated with Avg.C of CD8_4 neighboring with a tu_2 cell (r = 0.77, *p* = 5 × 10^−9^) and Avg.C of CD8_4 neighboring with a tu_3 cell (r = 0.72, *p* = 1 × 10^−7^). The feature elimination process of our logistic regression model first filtered out the highly correlated features as mentioned above (also see Methods). Owing to this elimination process, features that are highly correlated with any feature in the prognostic list identified by the logistic regression model may also have prognostic value. For example, because Avg.C of CD8_4 neighboring tu_1 was a prognostic feature, Avg.C of CD8_4 neighboring tu_2 or CD8_4 neighboring tu_3 may have similar prognostic values.

### 3.5. Correlations between Cell Subtype Density and Transcriptomic Profiles from Microdissected Fibroblastic Stromal and Epithelial Compartments of HGSC

To provide mechanistic insights by which certain cell phenotypes identified by IMC modulate survival in HGSC patients, we performed correlation studies and filtered genes by scRNA-seq [22] data (see Methods) to identify genes in the epithelial compartment (Figure 7A) or fibroblastic stromal compartment (Figure 7B) of HGSCs that had a significant positive or negative correlation with the IMC cell densities in the tumor tissue (Appendix A) and that were significantly different between LTS and STS (Appendix A). A total of 26 samples with whole transcriptome data available were used. We focused on the cell densities of the six IMC features significantly different between LTS and STS (Figure 5A), four of which were also selected by our machine learning model for survival prediction (Figure 6B, bottom). We discovered relationships consistent with known cancer biology while making unexpected observations.

In the epithelial compartment of HGSC tumors, we identified expression levels of several genes involved in the migration and invasion of cancer cells correlated with CD73_1, CD73_2, and CD31 densities and with poor survival, suggesting that these genes regulate the metastatic potential of cancer cells in a paracrine manner; these genes included *PARD6B*, *S100A10*, *SLURP1*, and *SPINT1*. In addition, we demonstrated that ovarian cancer cell–derived *ITIH5*, *TACSTD2*, and *WFDC2* expression levels were negatively correlated with granzyme B^+^ CD8^+^ cytotoxic T cell (CD8_4) densities. In contrast, *BPIFB1* and *SLURP1* expression levels were positively correlated, and *CD9* and *ITIH5* were negatively correlated with CD45RO^+^ CD4^+^ memory T cell (CD4_4) densities (Figure 7A and Appendix A). These findings suggest that these ovarian cancer cell–derived genes, which are known to code for extracellular matrix and metabolism modulators [34,35,36,37,38,39], could facilitate or block T cell infiltration in the tumor microenvironment as well as interfere with T cell activation and facilitate immune system evasion. Moreover, cancer cell–derived *FST* and *PARD6B* were positively correlated with CD31^+^ CD73^mid^ endothelial cell (CD31) density, indicating that these genes modulate endothelial cell activity and subsequently angiogenesis. The prognostic significance of these genes was determined by analyzing 530 samples of optimally debulked advanced HGSC in the KM Plotter, and the results are summarized in Appendix A.

To identify cancer-associated fibroblast (CAF)-derived mediators that confer the prognostic phenotypes identified by IMC, we analyzed correlations of the expression levels of genes in the transcriptomes generated from the microdissected fibroblastic stromal compartment of HGSC with the prognostic phenotypes identified. Survival analyses of the significantly correlated genes were also performed. The list of CAF-derived genes that were positively correlated with CD73^+^ cell (CD73_1) and CD73^mid^ cell (CD73_2) densities and also associated with STS is shown in Appendix A, Figure 7B, and Appendix A. These genes are likely expressed by CD73_1 and CD73_2, two CAF subtypes. Among these genes, *CCDC85B*, *DDAH1*, *EFEMP2*, *F2RL1*, *ITGB1*, *LOX*, *LDLR*, *MFGE8*, *MICAL2*, *MKL1*, *MSRB3*, *NCAM1*, *NPTX*, *PLAT*, *SLC2A3*, *SPSB1*, and *VASN* have been described as promotors of tumor cell growth, invasion, and migration, as well as angiogenesis [40,41,42,43,44,45,46,47,48,49,50,51,52,53,54,55] (Appendix A). *HMOX2*, *ICMT*, *MICU1*, and *TSPAN9* also positively correlated with CD73_1 and CD73_2 densities and short survival and have been shown to be involved in conferring chemoresistance in ovarian cancer [56,57,58,59] (Appendix A). CAF-derived *ANGPTL2*, *WWTR1*, and *PLOD2* promote tumor progression and cell invasion, and *MYO10* regulates CAF rigidity [60,61,62,63,64,65]. These findings indicate that these mediators produced by CD73_1 and CD73_2 CAF subtypes modulate malignant phenotypes of HGSC.

CAF-derived genes involved in the modulation of the immune response and associated with patient survival were examined. The results are summarized in Appendix A. Among these genes, *BMPR1B*, a gene encoding the receptor of the bone morphogenetic protein (BMP), was negatively correlated with CD4_4 cells and associated with STS. This finding suggests that CAFs expressing high levels of *BMPR1B* may be more responsive to BMP signaling, which subsequently modulates the rigidity of CAFs, stiffness of the tumor microenvironment, and CD4_4 T cell trafficking. Besides *BMPR1B*, *VSTM4*, a secreted protein that can reduce IFN-γ, IL-2, and IL-17 cytokine production by human T cells and cause a profound decrease in T cell activation [66], was negatively correlated with CD4_4 density but positively associated with STS, suggesting that CAF-derived VSTM4 modulates CD4_4 activity and subsequently leads to poor survival rates in patients with HGSC.

In addition to these correlations between expression of CAF-derived genes and prognostic immune cell phenotypes, expression of several genes with prognostic significance was also associated with the two CAF phenotypes CD73_1 and CD73_2. *FN1*, *TGFBI*, *TNC*, and *LRRC32* were positively correlated with both CD73_1 and CD73_2 and negatively correlated with survival. LRRC32 is a key regulator of TGF-β activation [67]. Together with increased TGFB1 secreted by CD73_1 and CD73_2 CAFs, LRRC32 may generate an immune-suppressive and pro-tumorigenic microenvironment to support the malignant phenotype of ovarian cancer cells, as we previously described [23,68]. Both *FN1* and *TNC* encode extracellular proteins that have previously been shown to be associated with short progression-free survival and increased migration-inducing potential in HGSC [69,70].

Several stromal genes were associated with density of endothelial cells (CD31). Among them, *MFGE8*, which was also positively correlated with CD73_1 and CD73_2 densities, demonstrated the strongest correlation with increased CD31 density and poor patient survival (Appendix A). Our findings are supported by published studies reporting that MFGE8 increases tumor angiogenesis by increasing VEGF and ET-1 expression in stromal cells and by enhancing M2 polarization of macrophages [46]. Moreover, MFGE8 proteins accumulated around CD31^+^ blood vessels have been shown to promote angiogenesis by enhancing PDGF-PDGFRβ signaling mediated by integrin-growth factor receptor crosstalk [47,71].

Taken together, the transcriptome analysis revealed several key genes in both cancer cell and fibroblastic stromal compartments of HGSC that correlate with IMC features positively or negatively. The presented SIO method indicates that these genes can promote either a tumor-promoting or an immune-suppressive microenvironment, subsequently leading to short-term survival in HGSC patients (Figure 7C).

## 4. Discussion

In the present study, we applied a new analytic pipeline, SIO, to location-specific, highly multiplexed IMC data based on deep learning and logistic regression to predict HGSC patients’ survival rates. We quantified the abundance of twenty-one markers in each sample, characterizing the immune milieu in both tumoral and stromal compartments. The advanced deep learning algorithm, MRCNN, incorporated in SIO has been shown to efficiently detect objects in an image while simultaneously generate a high-quality segmentation mask for each instance [19]. In this study, we demonstrated that the MRCNN algorithm can be adapted for cell segmentation in densely packed, structurally complicated, and widely varying IMC images with minor configuration. Our pipeline automatically split the image into tumor-enriched and non-enriched regions and identified prognostic cell density or nearest neighbor features in tumor-enriched regions that predict survival.

Accurate and reliable cell segmentation in tissues is always a challenging task, especially for large, highly multiplexed images such as IMC images. In 5 μm tissue sections, only a portion of a cell may be present within the analyzed section, many cells will overlap or do not have clear boundaries, and sizes and shapes of different cell types may vary a lot. Conventional microscopic image analysis algorithms, such as Watershed segmentation, has been applied to segment large, high resolution IMC images [17]. In this study, we showed that the deep learning algorithm outperformed watershed algorithm for IMC cell segmentation yet leveraging computerized outputs of watershed segmentation as the training sets to sidestep the laborious and time-consuming labeling efforts of training sets encountered in deep learning.

Recently, methods of spatial analysis on highly multiplexed imaging have been developed, such as neighbor analysis [15] and community analysis [16]. Neighbor analysis was reported to measure the significance of a neighbor interaction by permutation test [15]. In this study, we further extended the utility of neighborhood analysis to the nearest neighbor interactions in the tumor-enriched regions. To the best of our knowledge, this is the first study that integrates automated spatial feature selection and associated neighborhood analysis to predict clinical outcomes based on extracted specimens of cancer patients.

Spatial analysis of cell densities enables the identification of novel cell subtypes exhibiting significant differences between LTS and STS. In particular, the density of granzyme B^+^ CD8^+^ cytotoxic T cells (CD8_4) was significantly higher in LTS than in STS, expanding the prior knowledge that CD3^+^, CD4^+^, and CD8^+^ tumor-infiltrating lymphocytes are associated with positive outcomes in ovarian cancer [72,73,74,75] and highlighting that only a particular subtype of activated T cells (granzyme B^+^) is associated with good clinical outcomes in ovarian cancer. In addition, we found that the density of CD45RO^+^ CD4^+^ memory T cell (CD4_4) was significantly higher in LTS than in STS. Memory CD4^+^ T cells could provide a protective response against cancer cells by making effector cytokines respond early and by enhancing CD8^+^ T and B cell responses, as well as by secreting cytokines that can induce other cells in the tumor microenvironment to mount antitumor immunity [73].

Among different CD73^+^ cell subtypes, densities of CD73_1 and CD73_2, two different CAF subtypes, were significantly lower in LTS than in STS. CD73 is a GPI-anchored nucleotidase that catabolizes the production of extracellular adenosine and promotes tumor immune escape and thereby tumor growth. Indeed, CD73 expression has been shown to be associated with shorter disease-free and overall survival in HGSC patients and decreased CD8^+^ tumor-infiltrating lymphocytes [76,77]. A recent study reported that CAF-derived CD73 enforces an immune checkpoint [78]. These findings reinforce the hypothesis that CAFs play a role in shaping the immune landscape of the tumor microenvironment and modulating patient survival rates.

Among different tumor cell subtypes, B7H4^+^ Keratin^+^ tumor cell (tu_9) density was significantly lower in LTS than in STS. B7-H4 overexpression in cancer cells has been previously identified in high-grade ovarian tumors [79], but to the best of our knowledge, this is the first report showing that increased density of tu_9, a subtype of ovarian cancer cells expressing high levels of B7-H4, is associated with poor patient overall survival rates. Meanwhile, we found CD31^+^ CD73^mid^ endothelial cell (CD31) density was significantly lower in LTS than in STS. Although CD31 expression has shown no prognostic survival value, high CD31 expression was found in poorly differentiated tumors in a published study [80].

Cell-cell communication between heterogeneous tumor cells and various types of stromal cells, including infiltrating T cells, macrophages, CAFs, endothelial cells, and others, has been shown to be able to shape the ovarian cancer ecosystem, which in turn modulates disease progression and clinical outcome [3,74,81]. Our machine learning driven SIO pipeline identified ten nearest-neighbor, cell-cell interactions together with age as salient features to attain the best survival prediction accuracy of HGSC patients, with an AUC of 1. These findings indicate that cell-cell communication between different cell subtypes in the tumor microenvironment generates more reliable prognostic features in predicting patient survival rates than cell density alone. For example, CD31 neighboring tu_4 was negatively correlated with patient survival, confirming that increased angiogenesis supports tumor cell growth and leads to poor patient survival rates. In contrast, CD8_4 neighboring tu_1 was positively correlated with patient survival, suggesting that intratumoral activated CD8^+^ cytotoxic T cells closely interact with tumor cells as an attempt of an anti-tumor response and lead to improve patient survival rates, consistent with the findings reported in the literature [72,73,74,75]. Increased interaction between heterogenous populations of cell subtypes in the tumor microenvironment likely involves ligand-receptor crosstalk among different cell subtypes. Further experiments using spatially resolved single cell transcriptomes on one cell subtype with its nearest neighboring partner will be needed to validate the predicted cell-cell interactions as well as to understand how these interactions and the crosstalk signaling networks contribute to malignant phenotypes and their correlation with patient survival rates.

Several studies have coupled IMC data to multiplatform genomics to understand how the genome shapes the composition and architecture of tumor ecosystems [82,83]. To delineate the molecular mechanisms by which certain cell phenotypes identified by IMC modulate survival rates in HGSC patients, correlational studies have used genes in the microdissected epithelial compartment or fibroblastic stromal compartment of HGSCs associated with patient survival and prognostic cell phenotypes identified by IMC. We found that cancer cell–derived *WFDC2* negatively correlated with granzyme B^+^ CD8^+^ cytotoxic T cell (CD8_4) density and was associated with poor HGSC patient survival. In fact, *WFDC2*, which encodes the protein HE4, has been shown to correlate with poor survival in HGSC patients and promote tumor growth and confer chemoresistance in ovarian cancer [84]. Moreover, HE4 has been described as a driver of immune failure in ovarian tumors by compromising cytotoxic CD8^+^ T cells through upregulation of self-produced dual-specificity phosphatase 6 (DUSP6) [85]. These findings suggest that our SIO pipeline showed robust performance in identifying prognostic biomarkers associated with immune cell phenotypes described in previous studies. Besides *WFDC2*, we found that other relevant genes involved in the regulation of T cell activity, apoptosis, and infiltration were overexpressed in STS; including *ITM2B*, *ITIH5*, *CD9*, and *TACSTD2* [36,37,86].

CAFs have been shown to facilitate cancer progression by supporting tumor cell growth, extracellular matrix remodeling, angiogenesis, and formation of an immunosuppressive microenviroment [83]. These results are supported by our research showing that CAF subtypes, CD73_1 and CD73_2, express genes, such as *ANGPTL2*, *TNC*, *TGFB1*, *FN1*, *BMPR1B*, and *LRRC32*, promote tumor progression, angiogenesis, and extracellular matrix remodeling [60,61,67,69,70,87,88]. Some of these genes, including *TGFB1*, *BMPR1B*, and *LRRC32*, have been shown to modulate TGF-β signaling, which suppresses infiltration of anticancer immune cells such as cytotoxic T cells and natural killer cells and promotes the function of pro-cancer immune cells, such as regulatory T cells and M2 macrophages, in the tumor microenvironment [60,89,90], leading to poor patient survival rates. Using additional antibodies targeting these immune cell phenotypes in IMC analysis will further validate our observations.

## 5. Conclusions

In conclusion, the presented SIO analytic pipeline combined with transcriptomes generated from microdissected epithelial and fibroblastic stromal compartments of HGSC patient specimens demonstrates the heterogeneity of both tumor and stromal cell subtypes in HGSC. The spatial image-omics analysis also identified cellular features and phenotypes with prognostic significance and helped delineate the molecular mechanism by which these features modulate the tumor-promoting and immune-suppressive microenvironment (Figure 7C).

## Figures and Tables

**Figure 1 cancers-13-01777-f001:**
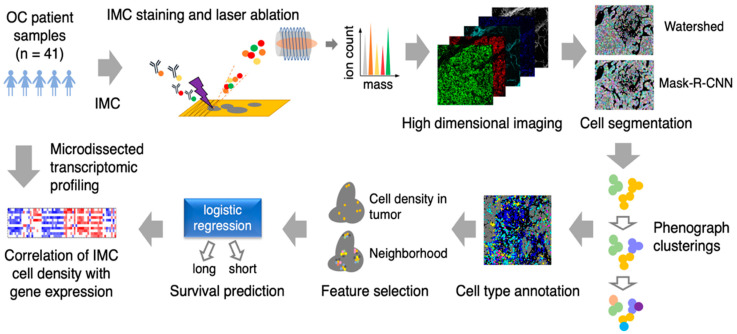
Schematic of imaging mass cytometry (IMC) acquisition of multiplexed images from 41 ovarian cancer patient samples and the spatioimageomics (SIO) pipeline. The SIO pipeline includes cell segmentation by a mix of the mask-R-CNN method trained on the outputs of the watershed method, phenograph clusterings, cell subtype annotation and visualizations, cell density and nearest-neighborhood feature selection, survival prediction by logistic regression, and integration of IMC cell density with transcriptomes from location-specific microdissected epithelial and fibroblastic stromal compartments of HGSC patient tissue samples.

**Figure 2 cancers-13-01777-f002:**
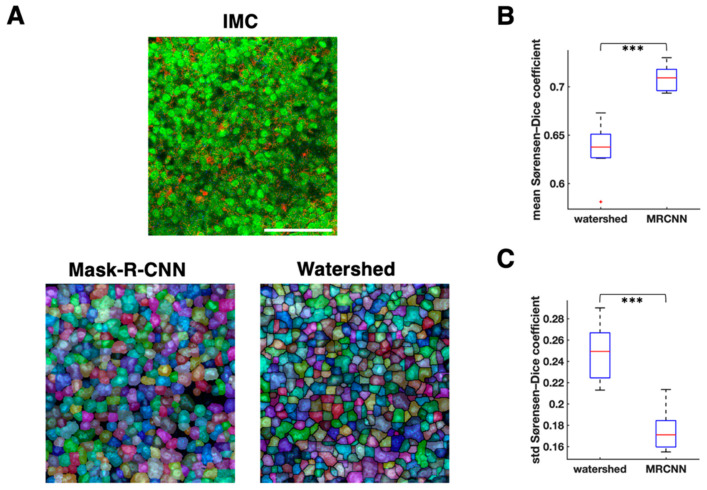
Comparison between watershed and mask-R-CNN cell segmentation. (**A**). Cell segmentation comparison on one representative image. Top, imaging mass cytometry (IMC) image; red: Keratin8/18, green: histone H3, blue: CD45. Bottom left, mask-R-CNN segmentation (transparent segmentations with pseudo colors are placed over nucleus image). Bottom right, watershed segmentation (transparent segmentations with pseudo colors are placed over nucleus image). White scale bar, 100 μm. (**B**). Mean Sørensen–Dice coefficient comparison between watershed and mask-R-CNN (MRCNN) with the reference segmentation drawn manually (*n* = 10 images, 300 cells segmented per image on average, paired *t* test). (**C**). Standard deviation of Sørensen–Dice coefficient comparison between watershed and MRCNN with the reference segmentation drawn manually (*n* = 10 images, 300 cells segmented per image on average, paired *t* test). *** *p* < 0.001.

**Figure 3 cancers-13-01777-f003:**
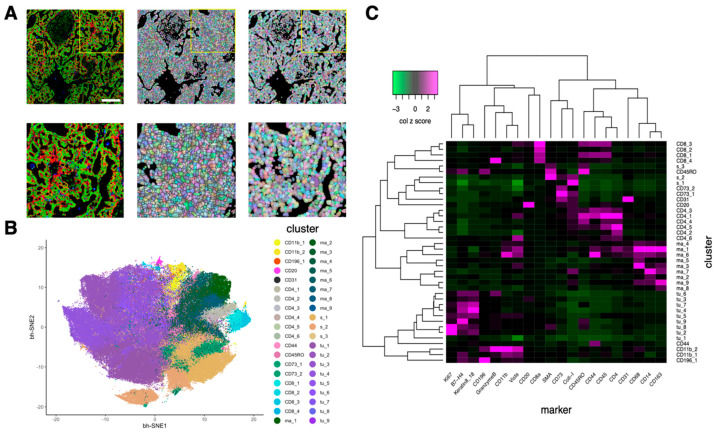
Cell segmentation by mask-R-CNN and cell subtype annotation based on marker expression. (**A**). Comparison of watershed segmentation and mask-R-CNN segmentation. Left, imaging mass cytometry (IMC) image; red: Keratin8/18, green: histone H3, blue: CD68. Middle, watershed segmentation. Right, mask-R-CNN segmentation. Bottom three images are the zoomed-in images of the yellow rectangle region of the top three images. White scale bar, 200 μm. (**B**). Two-dimensional bh-SNE representation of multiplexed IMC data highlighted by cell subtypes generated by phenograph clusterings. Each dot represents one cell. (**C**). Heatmap showing the median marker expression (z-scored by column) of each cluster (cell subtype). Twenty markers and 40 clusters were ordered by hierarchical clustering with the distance based on Pearson correlation.

**Figure 4 cancers-13-01777-f004:**
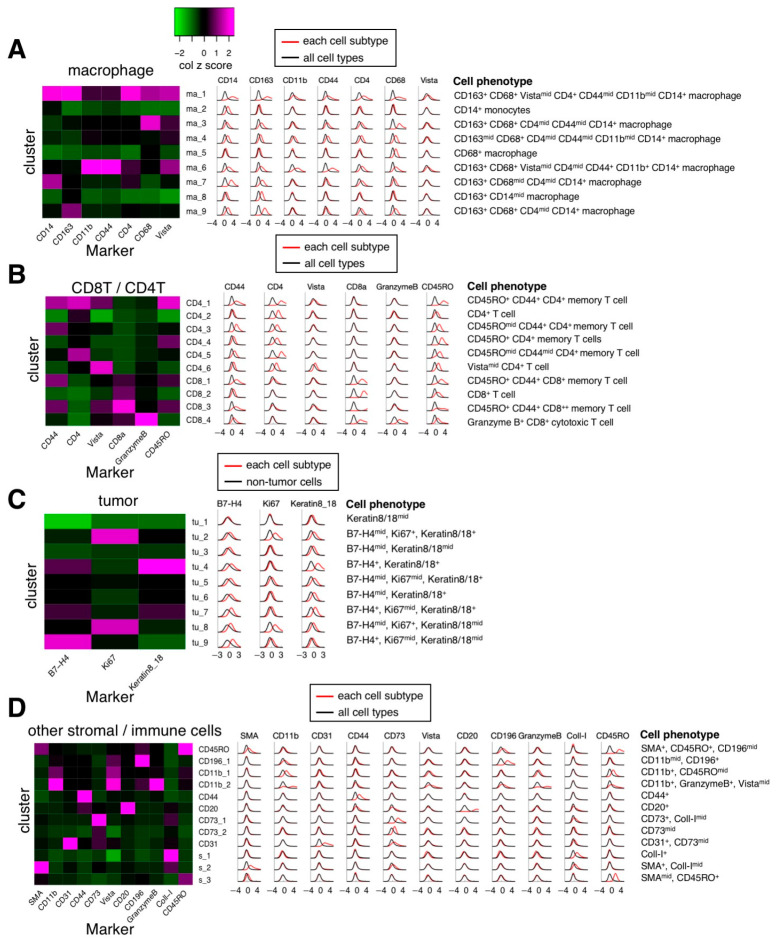
Marker expression levels of major individual cell subtypes. Left, heatmaps of median marker expression level (z-scored by column) for different major cell subtypes. Middle, histograms of Gaussian-smoothed (σ = 0.5) normalized marker expression levels of the cells of each cell subtype across all samples (red line) and normalized marker expression levels of the cells of all cell subtypes across all samples (black line, (**A**,**B**,**D**)) or normalized marker expression levels of the cells of non-tumor cells across all samples (black line, (**C**)). Right, summary of phenotypes of IMC cell subtypes. (**A**). Macrophage subtypes. (**B**). CD8^+^/CD4^+^ T cell subtypes. (**C**). Tumor subtypes. (**D**). Other immune cell subtypes.

**Figure 5 cancers-13-01777-f005:**
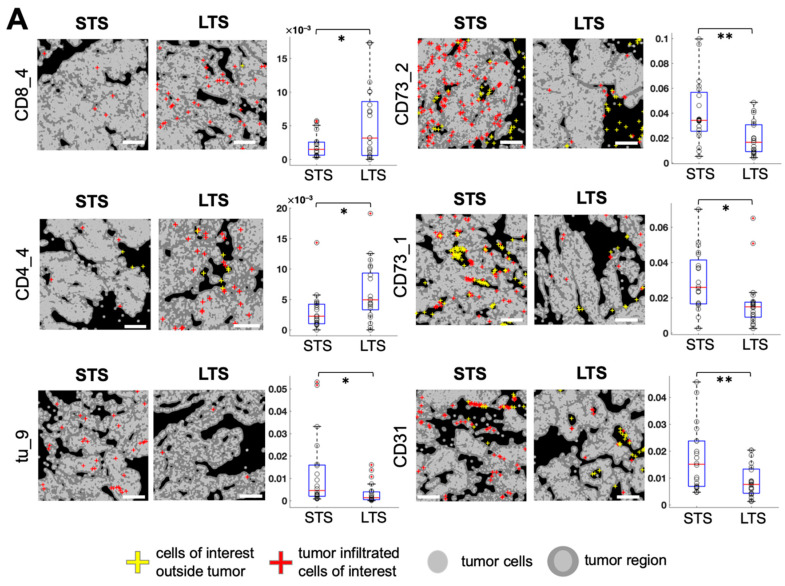
Identification of cell subtypes that exhibit density differences in tumor-enriched regions between long-term survivors (LTS) and short-term survivors (STS). (**A**). Left, visualization of the spatial distribution of tumor-infiltrated cells of interest in LTS and STS. White scale bar, 200 μm. Right, comparison of cell counts in tumor-enriched regions per total tumor cells for each cell subtype of interest between LTS and STS (*n* = 21 LTS, *n* = 20 STS, unpaired *t* test, * *p* < 0.05, ** *p* < 0.01). (**B**). Spearman correlation between cell densities of any two cell subtypes in the tumor-enriched regions. White color indicates correlation coefficient = 0 or *p* ≥ 0.05. Cell densities were ordered by hierarchical clustering with the Ward method.

**Figure 6 cancers-13-01777-f006:**
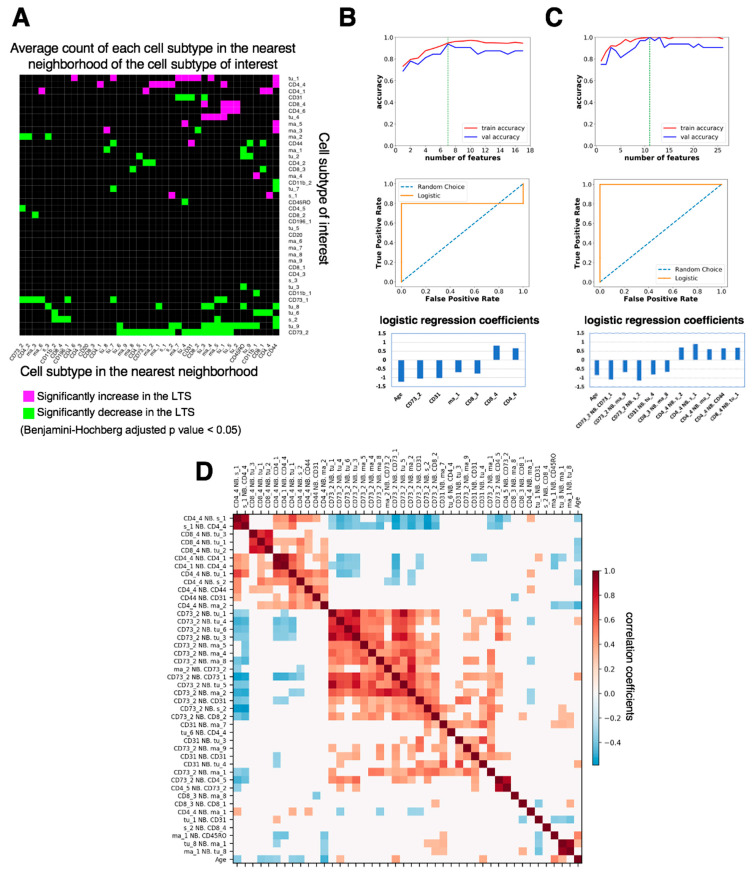
Comparison of nearest-neighbor cell-cell interactions between long-term survivors (LTS) and short-term survivors (STS) and feature selection for patient survival prediction. (**A**). Map of significant increases and decreases in nearest-neighbor interactions, computed as the average number of cell subtype X in the nearest neighbor of cell subtype Y (distance between the center of X and Y less than 20 μm), that are significantly (Benjamini-Hochberg adjusted *p* value < 0.05) increased (magenta) or decreased (green) in LTS compared with STS. (**B**). (**Top**) Number of cell density features selected by recursive feature elimination as a function of training (red) or validation (blue) accuracy. The optimal number of features is indicated by the green dashed line. Validation was done by leave-one-out cross validation. (**Middle**) Receiver operating characteristic (ROC) curve for the test set. (**Bottom**) Logistic regression coefficients of the features selected by the model. (**C**) (**Top**) Number of features of nearest-neighbor cell-cell interactions selected by recursive feature elimination as a function of training (red) or validation (blue) accuracy. The optimal number of features is indicated by the green dashed line. Validation was done by leave-one-out cross validation. (**Middle**) Receiver operating characteristic curve for the test set. (**Bottom**) Logistic regression coefficients of the features selected by the model. (**D**). Spearman correlation between the features that both correlate with patient survival (absolute correlation coefficient > 0.2). White color indicates correlation coefficient = 0 or *p* ≥ 0.05. Neighborhood features were ordered by hierarchical clustering with the Ward method.

**Figure 7 cancers-13-01777-f007:**
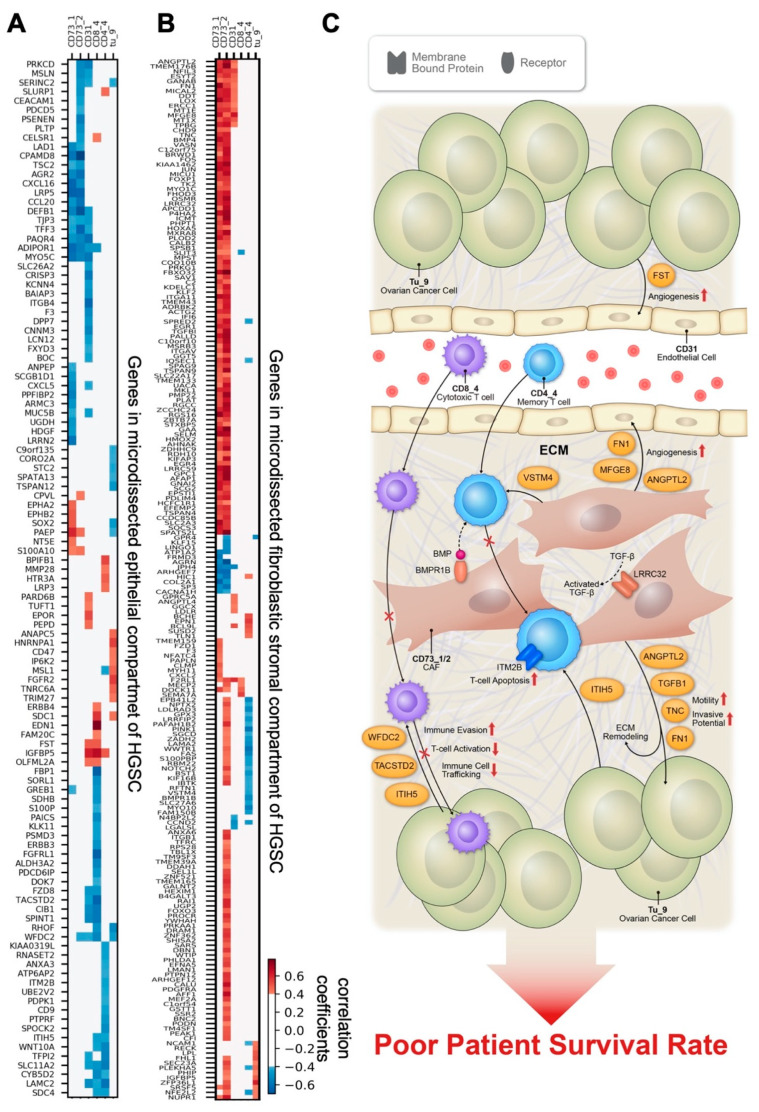
Gene correlation studies. Shown are correlations of gene expression in microdissected epithelial (**A**) and fibroblastic stromal (**B**) components of high-grade serous ovarian cancer (HSGC) samples with the imaging mass cytometry cell density in tumor-enriched regions (*n* = 26 samples, Spearman correlation, *p* < 0.05). White color indicates absolute correlation coefficient ≤0.4 or *p* ≥ 0.05. Genes were ordered by hierarchical clustering with the Ward method. (**C**) Schematic summarizing the genes that are significantly correlated with prognostic IMC features to postulate the mechanisms by which these genes contribute to the prognostic features. Cancer cell-derived WFDC2, TACSTD2, and ITIH5 drive immune surveillance failure by compromising cytotoxic CD8+ T cells activity and infiltration. CAF-derived VSTM4 can reduce cytokine production by CD4 T cells and cause a decrease in T cell activation. CAFs expressing BMPR1B are more responsive to BMP signaling, which subsequently modulates the rigidity of CAFs, and reduces CD4_4 T cell trafficking. CAF-derived LRRC32 suppresses anticancer immune cell infiltration by modulating TGF-β signaling networks. ANGPTL2, TNC, TGFB1, FN1, BMPR1B, and LRRC32 promote tumor progression, angiogenesis, and ECM remodeling.

## Data Availability

Raw data of the microdissected transcriptomes were downloaded from GEO with accession number GSE115635. The datasets/code supporting the current study are available from https://data.mendeley.com/datasets/4dpk7fjb58/draft?a=4aa42b24-f791-4a89-b74e-9dba795f3755, accessed on: 30 March 2021.

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
