# Peer review of "SIO: A Spatioimageomics Pipeline to Identify Prognostic Biomarkers Associated with the Ovarian Tumor Microenvironment"

_cancers, 2021, doi:10.3390/cancers13081777_

Round 1
Reviewer 1 Report
This is an interesting paper using machine learning approaches to characterize the role of the high grade serous ovarian cancer tumor microenvironment (TME), and the correlations between the TME composition and survival. The strengths are the detailed methodology and the integration of transcriptomics that provides mechanistic hypotheses.
The manuscript would be improved with additional information and approaches regarding the clinical specimens analyzed. Specific questions are as follows:
- Why was optimal cytoreduction defined as 1 cm given the increasing data the microscopic residual (R0) is central to clinical outcomes?
- The overall number of clinical specimens in this study is relatively small, with 41 patients - 21 long-term survivors and 20 short-term survivors. Why were these cohorts defined as overall survival of >= 60 months and <= 20 months? Were these group cutoffs pre-defined? Were patients with "intermediate" survival analyzed?
- Most importantly, given the small sample size, the separation of the training and test data sets at 80% and 20% leaves only 4-5 patients in the test set. The findings would be greatly strengthened with the inclusion of a larger cohort for the test set.
Reviewer 2 Report
Zhu et al present a thorough investigation of the tumor microenvironment in serous ovarian cancer patient samples. While previous work has examined specific aspects of the microenvironment, such as the impact of infiltrating CD8+ T cells on outcome, or the effect of tumor associated macrophages, this manuscript takes on the Herculean task of identifying subsets of each immune cell as well as endothelial cell and CAF markers to create a more complete picture of the microenvironment. Furthermore, the group chooses to analyze the data in multiple ways, using not only the density of cell types within the tumor, but also nearest neighbor and cell-cell interaction analyses to tease apart the interacting components of the tumor microenvironment that are implicated in long term versus short term survival. Finally, the authors compare their MRCNN analysis with RNA-Seq of microdissected tumor samples. This is a well written paper that in particular, provides additional information concerning the subsets of immune cells existing within the ovarian cancer microenvironment and how these subsets impact survival. This is an important study that warrants publication.
For the transcriptomics profiling, I was surprised that immune signatures weren’t examined as well as cancer vs stromal signatures, since the immune components appeared to be the most different between long and short term survivors. This has been done with other bulk RNA-Seq datasets. For instance, Finotello et al. Molecular and pharmacological modulators of the tumor immune contexture revealed by deconvolution of RNA-seq data. Genome Med 11, 34 (2019). You might comment on why this was not possible to examine in this data.
Minor issues:
For Figure 5A, it is difficult to see the yellow circles designating “cells of interest”. The majority of red crosses appear to be lacking a yellow circle. The yellow circle should be included for all crosses, or one could simply replace the yellow circle with a yellow cross with yellow crosses designating cells of interest outside of the tumor area and red crosses, tumor infiltrated cells of interest.
Figure 6, the figures should be labeled B and C without the additional a,b,c. The description can be top, middle and bottom graphs perhaps?
